# Variation of Microbial Community and Fermentation Quality in Corn Silage Treated with Lactic Acid Bacteria and *Artemisia argyi* during Aerobic Exposure

**DOI:** 10.3390/toxins14050349

**Published:** 2022-05-17

**Authors:** Weiwei Wang, Zhongfang Tan, Lingbiao Gu, Hao Ma, Zhenyu Wang, Lei Wang, Guofang Wu, Guangyong Qin, Yanping Wang, Huili Pang

**Affiliations:** 1School of Physics and Microelectronics, Zhengzhou University, Zhengzhou 450052, China; bingzhi213608@163.com (W.W.); mahaoworks@foxmail.com (H.M.); 2Henan Key Lab Ion Beam Bioengineering, School of Agricultural Sciences, Zhengzhou University, Zhengzhou 450052, China; tzhongfang@zzu.edu.cn (Z.T.); wang0708zy@outlook.com (Z.W.); qinguangyong@zzu.edu.cn (G.Q.); wyp@zzu.edu.cn (Y.W.); 3School of Biological and Food Engineering, Anyang Institute of Technology, Anyang 455000, China; gulingbiao@foxmail.com; 4Academy of Animal Science and Veterinary Medicine, Qinghai University, Xining 810016, China; wanglei382369@163.com (L.W.); jim963252@163.com (G.W.)

**Keywords:** whole crop corn silage, additives, fermentation characteristics, microbial communities, mycotoxin, total flavonoid

## Abstract

Silage, especially whole crop corn silage (WCCS), is an important part of ruminant diets, with its high moisture content and rich nutrient content, which can easily cause contamination by mold and their toxins, posing a great threat to ruminant production, food safety and human health. The objective of this study was to examine effects of lactic acid bacteria (LAB) *Lactiplantibacillus* (*L.*) *plantarum* subsp. *plantarum* ZA3 and *Artemisia argyi* (AA) on the fermentation characteristics, microbial community and mycotoxin of WCCS during 60 days (d) ensiling and subsequent 7 d aerobic exposure. The results showed that WCCS treated with LAB and AA both had lower pH value and ammonia nitrogen (NH_3_-N) contents, and higher lactic and acetic acids concentration compared with other groups after 60 d ensiling. In addition, for microbial communities, *Acetobacter* and *Enterobacter* were inhibited in all AA group, while higher abundance of *Lactobacilli* was maintained; besides, *Candida*, *Pichia* and *Kazachstania* abundances were decreased in both 6% and 12% AA groups. The content of five kinds of mycotoxins were all significantly lower after 7 d of aerobic exposure. As for the total flavonoid (TF), which is significantly higher in all AA treated groups, it was positively correlated with *Paenibacillus*, *Weissella* and *Lactobacilli*, and negatively with *Acetobacter*, *Enterobacteria*, *Kazachstania* and *Pichia*.

## 1. Introduction

Whole crop corn silage (WCCS) is the most common forage fed to ruminants because of nutritional value and high comprehensive economic benefits, but it can be contaminated with a variety of mycotoxins before harvest, during fermentation and after silage is completed. Especially aerobic deterioration and toxin production under aerobic conditions [1], resulting in a loss of up to 30% of the dry matter (DM) in nutrients [2], as well as mycotoxin, kind of secondary metabolities produced by different fungi species, which could remain in fermented feed even after mold themselves have disappeared, are known for their harmful effects on animal and human health, such as reducing animal performance, decreasing feed intake, increasing incidence of disease and impairment of the immune system [3,4,5]. Therefore, it is imperative and meaningful to inhibit secondary fermentation, improve aerobic stability and reduce mycotoxin of WCCS for making high-quality and safe silage.

Secondary fermentation of silage is closely related to the variety and moisture content of the material, preparation method, density and changes in ambient temperature, and is usually the result of a combination of factors [6]. In order to prevent the occurrence of secondary fermentation, physical methods, chemical treatments, and the addition of enzymes and probiotics are widely used in research or production to reduce the chance of secondary fermentation of silage, to improve the aerobic stability of silage and to maintain the quality of silage [7,8,9]. Of which, the microbial preparations, mainly lactic acid bacteria (LAB) (homo- and hetero-fermentation), ensure the fermentation quality and stability of the silage by regulating the microbiota in the silage raw material and making it possible to preserve its nutrient content. However, there is still a long way to go to study the effect of the amount and proportion of bacterial additive on the nutritional quality, fermentation characteristics and microbial population of silage during the aerobic stabilization phase, and the pattern of bacterial additive is not able to balance the requirements of silage fermentation quality and spoilage inhibition after aerobic exposure [10,11,12]. Therefore, it is important to develop new silage additives that can effectively maintain the anaerobic environment of forage during silage extraction, control aerobic stability and inhibit secondary fermentation to ensure silage quality.

Many countries have begun to forbid animal feed containing antimicrobial agents because of environmental pollution and residual antibiotics. The relevant authorities of China has ordered to effectively replace antibiotics with organic acids, probiotics, prebiotics, minerals and herbs. *Artemisia argyi* (AA) is a traditional Chinese herb medicine used for treating diseases such as diarrhea, hemostasis and inflammation, and is one of the most popular plant in China and eastern Asia [13]. Studies have proved that AA contains varieties of active compounds such as essential oils, polysaccharides, flavonoids, phenols, terpenoids and glycosides, and also contains crude protein (CP), polyunsaturated fatty acid, vitamin C and essential amino acids [14,15]. Flavonoids, which improve the antioxidant capacity of silage, reduce pH, ammonia nitrogen (NH_3_-N) content and the number of undesirable microorganism during fermentation [16,17], acting as free radical scavengers and metal chelators in the organism [18], performing functions such as bactericidal, boost immunity, antioxidant, antiviral and cardiovascular regulation [19]. It has been reported that AA had strong antibacterial effects against *Staphylococcus aureus*, *Escherichia coli* and *Salmonella enteritidis*, and showed anti-fungi activity [20,21]. In recent years, AA has been actively applied as a feed additive in animal production. Zhang et al. [22] found that dietary AA could increase antioxidant capacity of the serum and reduce the MDA content of broilers; Liu et al. [23] used AA as the dietary supplementation that could decrease the diarrhea rate and diarrhea index, and increase the small intestine length villus height of rabbits. In addition, AA also has a unique aroma that can fend off mosquitoes, flies and mites to improve the breeding environment and increase palatability and feed intake of diets in animals [24]. In addition, most of AA are distributed in wilderness, grassland and roadside. Therefore, the use of AA with many biologically active compounds as feed additives not only enhances animal production, but lowers feed cost. However, no information is related to AA as a feed additive, especially in WCCS.

At present, there has been no proper solution to the important problem of secondary fermentation in the practical application of WCCS. In addition, there is hardly any research on the effects of AA as an additive on the silage. Thus, the aim of this study was to compare the effects of adding AA with LAB on the fermentation quality, microbial flora and mycotoxin of WCCS during 60 days (d) fermentation and subsequent 7 d aerobic exposure. Through the study of the inhibition of fungi and their derivative mycotoxin by LAB and AA, we provide theoretical support for the production of high quality silage.

## 2. Results

### 2.1. Characteristics of Whole Plant Corn and Artemisia Argyi before Ensiling

The characteristics and microbial population of whole crop corn (WCC) and AA are shown in Table 1. WCC was harvested at the milk stage, and had a pH value of 6.61, while AA had a pH value of 5.58. The DM concentration for WCC and AA were 277.03 and 371.55 g/Kg, and WSC (water-soluble carbohydrate), CP (crude protein), NDF (neutral detergent fiber), ADF (acid detergent fiber) and EE (ether extract) contents were 12.12, 9.43, 60.07, 35.66, 2.51 (%DM) for WCC, and of AA were 5.41, 10.92, 53.76, 33.42, and 3.06 (%DM), respectively. However, relatively high counts as 4.68–8.32 colony forming units (cfu)/g fresh matter (FM) of undesirable microorganisms, including aerobic bacteria, coliform bacteria, yeast and bacilli, were observed on WCC.

### 2.2. Fermentation Quality, Chemical Composition and Microbial Population of Silage

#### 2.2.1. Fermentation Quality

Table 2 showed the significant effects of period, treatment and interactions on pH, organic acids, NH_3_-N and WSC (*p* < 0.05). The pH value of each treatment group decreased significantly after 60 d of silage compared with WCC (*p* < 0.05); after aerobic exposure for 3 d, all treatments increased remarkably except 12% AA (*p* < 0.05); and all treatments still continued to increase after aerobic exposure for 7 d, while 12% AA remaining had the lowest pH of 4.64. After 3 d of aerobic exposure, lactic acid content in all groups was reduced except 12% AA, and it was only detected in 6% and 12% AA after aerobic exposure of 7 d. At 60 d of ensiling, WSC contents were all significantly reduced compared with WCC. NH_3_-N showed an upward trend with the prolonging of aerobic exposure time, and CK silage had the highest content both at 3 and 7 d of aerobic exposure (*p* < 0.05), were 21.12% and 41.09%, respectively.

#### 2.2.2. Chemical Composition

As shown in Table 3, there was significant effect of period × treatment interactions found on silage CP, NDF, ADF (*p* < 0.01) and EE (*p* < 0.001). There were no differences in DM contents between CK and ZA3, but they were significantly lower than silages treated with AA at 60 d silage and aerobic exposure phase (*p* < 0.05). The CP of all silages showed no significant difference after 60 d of fermentation (*p* > 0.05), but it decreased with prolonged aerobic exposure time, while the loss of CP in 12% AA was the least compared with other groups (*p* < 0.05). Four treatments with AA significantly affected the ADF and EE contents (*p* < 0.05) after 60 d ensiling, and their levels after exposure to air were not very high.

#### 2.2.3. Microbial Population

Effect of period, treatment, and their interaction on count of LAB, aerobic bacteria, coliform bacteria, yeast and bacilli are listed in Table 4. The number of LAB increased after fermentation of 60 d compared with WCC, all counts of LAB were significantly lower at aerobic exposure stage (*p* < 0.05), especially after 7 d of aerobic exposure LAB was not detected except 6% and 12% AA. The number of undesirable bacteria including aerobic bacteria, coliform bacteria, yeast and bacilli in each group followed a decreasing trend after 60 d of fermentation, while the number of them increased dramatically after aerobic exposure, and both reached a peak after 7 d of aerobic exposure. Among which, the lowest levels of undesirable bacteria were maintained in 12% AA (*p* < 0.05).

### 2.3. Diversity of Bacteria and Fungi

Alpha diversity of bacterial and fungi in WCCS are presented in Table 5. For bacteria, Shannon indexes showed that lower diversity was observed in 6% and 12% AA compared with other groups at aerobic exposure 7 d (*p* < 0.05), which were 4.31 and 4.53, respectively, and significantly higher than 60 d ensiling (*p* < 0.05); Chao1 indexes were decreased in 6% and 12% AA groups after exposure 3 d compared to 60 d ensiling (*p* < 0.05), at 1816.18 and 1886.84, respectively, and were also significantly lower in these two groups than in CK at 7 d of exposure (*p* < 0.05).

Both the Chao1 and Shannon indexes of the fungi community in 6% and 12% AA groups were lower than other groups after aerobic exposure for 7 d (*p* < 0.05), and at this point, Chao1 indexes of these two groups reached the minimum throughout the whole experiment with 65.33 and 54.61, respectively, while there was no significant change in Shannon indexes between these two groups compared to 60 d of silage.

As shown in Figure 1, distinctions of bacterial and fungi community among six groups were clear. For bacterial community, CK were separated apparently from those of treated groups after 60 d of ensiling, and the structure of the bacteria had changed a lot after aerobic exposure, the variation became more obvious with the aerobic exposure time prolonged. CK and ZA3 groups were more similar at 7 d aerobic exposure (Figure 1A–C). As to fungi community, the CK, 6% and 12% AA+ZA3 silage were similar on day 60 ensiling. However, after aerobic exposure, a clear separation and difference among six treatments, and clustered in four quadrants was observed, respectively (Figure 1D–F).

### 2.4. Relative Abundances of Bacterial and Fungi

The relative abundance on phylum and genus levels of bacterial and fungi communities in WCCS are elucidated in Figure 2. 

#### 2.4.1. Phylum Level

At the phylum level, for bacteria (Figure 2A), after 60 d fermentation, Firmicutes, which in CK was 57.75, and in treated groups were 80.58, 93.12, 87.13, 57.64 and 58.25%, respectively, was the dominant bacteria in all groups. In addition, after 60 d fermentation, 6% AA+ZA3 was also dominated by Bacteroidetes (17.33%), Actinobacteria (16.34%) and Fusobacteria (11.63%), and in 12% AA+ZA3 were 13.17, 10.78 and 7.61%, respectively. After 3 d of aerobic exposure, Proteobacteria had replaced Firmicutes and became the dominant phylum, which in CK was 69.75%. Additionally, 6% AA addition reduced the abundance of Proteobacteria, while ZA3 did the opposite; the abundance of Bacteroidetes, Actinobacteria and Fusobacteria in 6% AA+ZA3 were 6.02, 5.06 and 3.02%, in 12% AA+ZA3 were 16.99, 18.26 and 9.27%, and in 12% AA were 7.81, 5.14 and 0.89%. As for after 7 d aerobic exposure, Firmicutes were rapidly decreased in CK, ZA3, 6% AA, 12% AA and 6% AA+ZA3 groups with 21.25, 16.06, 64.50, 68.64, and 46.62%, respectively, while in 12% AA+ZA3 also with 56.32%; all groups showed further increases of Proteobacteria, especially in ZA3 was 81.76%, which also replaced Bacteroidetes, Actinobacteria and Fusobacteria in 6% AA+ZA3, 12% AA+ZA3 and 12% AA.

After 60 d of silage, the dominant fungi on the phylum level were Ascomycota in all groups (Figure 2B), which in CK was 80.10, and in treated groups were 92.85, 94.75, 85.99, 93.54 and 94.41%, respectively. After aerobic exposure for 3 d, the abundance of Ascomycota increased in all groups; they were 91.99, 95.00, 99.89, 99.95, 96.99 and 97.87%, respectively. While aerobic exposure for 7 d, Ascomycota in ZA3, 6% and 12% AA+ZA3 reduced to 90.98, 94.96 and 94.00%, respectively, and other groups maintained above 98.90%.

#### 2.4.2. Genus Level

As at the genus level, for bacteria (Figure 2C), after 60 d fermentation, the structures of the flora in CK and treated groups were significantly different. *Lactobacilli* in CK was 37.11%, while in treated groups it was 77.14, 74.84, 68.74, 48.40 and 42.38%, respectively. *Enterobacteria* was 5.16% in CK, while all under 0.27% in treated groups. In addition, the relative abundance *Pediococcus* and *Flavobacterium* were higher in CK and ZA3, were 9.90 and 8.53% in CK, 8.01 and 3.52% in ZA3, while *Weissella* were higher in 6% (2.11%) and 12% AA (7.66%). After aerobic exposure for 3 d, a relatively high abundance of *Lactobacilli* was found in 6% (71.26%) and 12% AA (70.08%), and other four groups were reduced to 27.95, 56.76, 33.77 and 37.13%, respectively, consistent with the decrease of lactic acid after aerobic exposure, while *Enterobacter* increased from 16.87% to 28.58% in CK, and no significant changes was observed in treated groups. In general, *Acetobacter* became apparent after aerobic exposure, which destroyed fermentation quality and aerobic stability of silages. In this study, *Acetobacter* was 47.64 in CK, and was 10.02, 10.33, 0.01, 54.43 and 40.02% in treated groups, respectively. This might be due to the presence of acetic acid in 12% AA, which inhibited the growth of *Acetobacter* after 3 d of aerobic exposure. After aerobic exposure 7 d, *Lactobacilli* significantly decreased in all groups, in CK was 3.83, and in treated groups was 5.59, 31.12, 36.88, 5.36 and 7.59%, respectively. While *Enterobacter* in CK, ZA3, 6% and 12% AA+ZA3 increased to 15.25, 10.01, 5.01 and 7.30%, respectively. Meanwhile, it was undetectable in 6% and 12% AA. *Acetobacter* was also dominant in all groups after aerobic exposure 7 d, which was 51.36 in CK, while 37.78, 22.34, 26.80, 36.92 and 32.08% in treated groups, respectively. In addition, after aerobic exposure 7 d, *Bacillus* and *Paenibacillus* increased by adding AA, *Bacillus* in 6% AA, 12% AA, 6% and 12% AA+ZA3 were 9.43, 8.34, 18.68 and 22.51%, respectively, while in CK and ZA3 were 5.65 and 1.68%. *Paenibacillus* was 6.96, 6.19, 6.41 and 13.57% in 6% AA, 12% AA, 6% and 12% AA+ZA3, respectively, while in CK and ZA3 it was 1.25 and 4.16%.

For fungi (Figure 2D), *Kazachstania*, *Candida* and *Pichia* were dominated in WCCS after 60 d fermentation and aerobic exposure. After 60 d of silage, the relative abundance of *Kazachstania* was higher in CK, ZA3, 6% and 12% AA+ZA3 with 42.25, 55.46, 43.49 and 40.83%, respectively, while it was lower in 6% (31.77%) and 12% AA (29.11%); *Pichia* in CK and 6% AA+ZA3 was 14.11 and 13.11%, but in other groups all under 4.49%; *Candida* in CK was 35.81, and in treated groups was 19.88, 16.29, 24.80, 31.19 and 45.66%, respectively. After aerobic exposure 3 and 7 d, *Kazachstania* in 6% and 12% AA were all lower than that in other groups, *Pichia* in ZA3 increased while 6% AA+ZA3 decreased, *Candida* was the highest in CK and was the lowest in 12% AA. 

### 2.5. Correlation Analyses of the Bacterial and Fungi Community with Fermentation Properties

Fermentation and aerobic exposure are complex processes of interaction between fermentation products and microbial communities. Figure 3 illustrates the relationships among the top 10 most abundant of bacterial or fungi genera and fermentation properties. At ensiling 60 d (Figure 3A,D), pH was positively correlated with *Weissella* (*r* = 0.72), *Pediococcus* (*r* = 0.77), *Lactococcus* (*r* = 0.75), *Paenibacillus* (*r* = 0.75) and *Naumovozyma* (*r* = 0.75), and negative with *Candida* (*r* = −0.52) and *Tetrapisispora* (*r* = −0.70), while acetic acid was positively with *Weissella* (*r* = 0.83), *Pediococcus* (*r* = 0.48), *Lactococcus* (*r* = 0.77), *Paenibacillus* (*r* = 0.58) and *Naumovozyma* (*r* = 0.51), and negative with *Candida* (*r* = −0.50), *Tetrapisispora* (*r* = −0.69) and *Wallemia* (*r* = −0.50). NH_3_-N was positively with *Weissella* (*r* = 0.69), *Pediococcus* (*r* = 0.70), *Lactococcus* (*r* = 0.71), *Pichia* (*r* = 0.54), *Naumovozyma* (*r* = 0.71) and *Coprinopsis* (*r* = 0.48), and negative with *Tetrapisispora* (*r* = −0.72), and WSC was positively with *Enterobacter* (*r* = 0.48), *Pichia* (*r* = 0.58) and *Penicillium* (*r* = 0.51).

After aerobic exposure for 3 d (Figure 3B,E), pH was positively with *Kazachstania* (*r* = 0.48) and *Pichia* (*r* = 0.50), and negative with *Lactobacilli* (*r* = −0.68), lactic acid was negative with *Kazachstania* (*r* = −0.60) and *Tetrapisispora* (*r* = −0.52). Additionally, acetic acid was positively with *Lactobacilli* (*r* = 0.79) and *Bacillus* (*r* = 0.54), and negative with *Kazachstania* (*r* = −0.83) and *Pichia* (*r* = −0.51), NH_3_-N was positively with *Weissella* (*r* = 0.61), *Pediococcus* (*r* = 0.68) and *Acetobacter* (*r* = 0.62), while WSC was positively with *Flavobacterium* (*r* = 0.48). Genus *Flavobacterium* was barely reported in silage, and requires further study. After aerobic exposure for 7 d (Figure 3C,F), pH was positively with *Enterobacter* (*r* = 0.81), *Kazachstania* (*r* = 0.62) and *Pichia* (*r* = 0.69), and negative with *Lactobacilli* (*r* = −0.66), while lactic acid was positively with *Lactobacilli* (*r* = 0.88), *Pediococcus* (*r* = 0.59), *Acetobacter* (*r* = 0.64) and *Lactococcus* (*r* = 0.61), and negative with *Enterobacter* (*r* = −0.60), *Kazachstania* (*r* = −0.59) and *Tetrapisispora* (*r* = −0.47). NH_3_-N was positively with *Enterobacter* (*r* = 0.73) and *Pichia* (*r* = 0.75), and negative with *Lactobacilli* (*r* = −0.55) and *Lactococcus* (*r* = −0.47). This indicates that these microorganisms might obstruct the production of NH_3_-N or be inhibited by NH_3_-N.

### 2.6. Toxin Content

The content of toxin during 60 d fermentation and aerobic exposure are shown in Figure 4. After 60 d fermentation, the Aflatoxin B1 (AFB1), deoxynivalenol (DON), zearalenone (ZEN), ochratoxin (OA) and fumonisin (FUM) in all groups were below the maximum content in feed. As for aerobic exposure 3 d, all contents of 5 mycotoxins in the CK were increased compared with 60 d of fermentation, and had significant in OA and FUM (*p* < 0.05). For aerobic exposure 7 d, AFB1, DON, ZEN, OA and FUM in CK reached again to 82.76 μg/kg, 2.56 mg/kg, 4.45 mg/Kg, 275.46 μg/kg and 29.14 mg/kg DM, respectively, and all exceeded the maximum content in feed; the same phenomenon existed in groups ZA3, 6% and 12% AA+ZA3, while the AFB1 in 12% AA, DON, ZEN, OA and FUM in 6% AA were the lowest, respectively, and the content of all five mycotoxins in groups 6% and 12% were lower than maximum content in feed.

### 2.7. Total Flavonoid and Linkages with Microbial

TF were evaluated for 60 d fermentation and aerobic exposure in Figure 5A. The contents of TF in 6% AA, 12% AA, 6% and 12% AA +ZA3 were significantly higher than that in CK and ZA3 groups after 60 d fermentation with 11.02, 11.38, 12.09 and 14.65 mg/g DM, and 7.72 and 8.54 mg/g DM in CK, ZA3 groups, respectively (*p* < 0.05). After aerobic exposure for 3 d, the content of TF had no significant different in CK, ZA3, 12% AA and 6% AA+ZA3 compared with 60 d fermentation, while it decreased to 10.09 and 12.12 mg/g DM in 6% AA and 12% AA +ZA3 (*p* < 0.05), respectively; while exposed for 7 d, it decreased significantly in CK, ZA3, 6% and 12% AA+ZA3 (*p* < 0.05), but had no significant changes in 6% and 12% AA (*p* > 0.05). 

Linkages between microbial and TF are shown in Figure 5B–D. After 60 d fermentation, TF were positive with *Tetrapisispora*, and negative with *Pediococcus*, *Acetobacter*, *Enterobacteria* and *Pichia*; and positive with *Tetrapisispora*, *Paenibacillus* and *Weissella*, and negative with *Pediococcus*, *Acetobacter* and *Enterobacteria* after aerobic exposure for 3 d; as for 7 d exposure, TF were positive with *Paenibacillus* and *Lactobacilli*, and negative with *Enterobacteria*, *Pichia*, *Kazachstania* and *Flavobacterium*.

## 3. Discussion

Generally, the fermentation quality is largely influenced by the characteristic of the raw material and the microorganisms epiphytic on its surface [25]. In this study, the WSC content in WCCS beyond 6% DM with a strong fermentability. It is reported that greater than 5.00 lg cfu/g FM LAB at ensiling is necessary for high-quality silage feed [26]. The populations of LAB on WCC and AA were 6.33 and 5.59 lg cfu/g FM, which was enough for initiating lactic acid fermentation under anaerobic condition, and epiphytic LAB could convert WSC into organic acids and decrease pH. However, higher amounts of undesirable microorganisms as aerobic bacteria, coliform bacteria and bacilli were also observed in both of the two raw materials at more than 4.68 lg cfu/g FM, which might be a challenge to ensile directly without additives. Therefore, to inhibit harmful microorganisms and enhance fermentation, it is necessary to add exogenous additives.

With prolonged aerobic exposure time, environment changed from anaerobic to aerobic, yeast in the silage proliferated rapidly, which consumed a lot of lactic acid, and caused pH to increase [27]. In this study, after aerobic exposure for 3 d, pH value in all treatments increased remarkable except 12% AA (*p* < 0.05), and all still continue increased after aerobic exposure for 7 d. In addition, the content of acetic acid was detected in CK, ZA3, 6% and 12% AA except 6% and 12% AA+ZA3 after 60 d of silage; and it sharply disappeared except 12% AA during the aerobic exposure, but the pH kept the same in 12% AA. This might speculate due to 12% AA produced organic acids and lowered the pH in the early stage of aerobic exposure, but with extended of aerobic exposure, the number of fungi increased and produced alkaline substances neutralizing part of acids, while only acetic acids was left. As a weak acid, acetic acid not sufficient to lower pH, but could significantly inhibit fungi and improve aerobic stability of silage [28,29]. Similar changes were also found in the study of Wang et al. [30], on d 1 and 2 of aerobic exposure, *L*. *plantarum*, *L*. *hilgardii* and their combination significantly increased the content of total acids, but remained the same pH value. Based on these, 12% AA might increase aerobic stability by producing acetic acid as a means to inhibit yeast growth during aerobic exposure stage. 

Protein in silage often hydrolyzed into non-protein nitrogen like ammonia and peptides by microbial activity and proteases, and NH_3_-N reflect the degree of peptide bond hydrolysis and amino acid deamination [31]. NH_3_-N showed an upward trend with the prolonging of aerobic exposure time, which might be due to the presence of various microorganisms; while the lowest was detected in 12% AA, which might be because lower pH inhibited the growth protein-hydrolyzing microorganisms. Additionally, attributed to the decreased pH values and inhibited the activities of undesirable bacteria in AA silages, DM concentrations in all AA groups were higher than WCC [32,33]. After aerobic exposure for 7 d, the content of NDF was significantly decreased compared to 60 d fermentation (*p* < 0.05), which might contribute to the number of *Bacillus* significantly increased in all groups producing cellulase or other enzymes to degrade NDF [34]. 

The CP and EE for AA raw material were 10.92 and 3.06 (%DM), and of WCC were 9.43 and 2.51 (Table 1), respectively. It can be seen that these two indicators in AA and WCC have no significant difference (*p* > 0.05). However, during the whole aerobic exposure phase, CP in all AA treated groups were higher than that in CK (Table 3), while the loss of CP in 12% AA was the least (*p* < 0.05) and in line with the change of NH_3_-N. This probably due to proteins are degraded by microorganisms into short-chain peptides, free amino acids and non-protein nitrogen such as ammonia, and these products can be used by microorganisms to synthesize microbial proteins, thereby increasing the true protein content of AA added silage [16]. The similar observation was found by Hu et al. [33] in corn silage. As for EE, which was remained higher in all AA added groups than CK from ensiling of 60 d until aerobic ends, might also contributed to beneficial bacteria consume part of the organic material by respiration during the fermentation process, thus releasing CO_2_ and H_2_O and reducing the total amount of products, resulting in a “concentration effect”; even more, a black oily substance that was presumed to be the volatile oil in all AA treated groups was observed on the measuring cups of SER148 EE analyzer during EE testing, while CK and ZA3 were transparent in contrast, which is speculated that the fermentation might have facilitated the breakdown of the large molecules in the AA into volatile oil.

Due to the proliferation of yeast, and typically the consumption of organic acids and sugars, which directly result in an increase in ambient temperature and pH value, and the fatty acids produced by the yeast might have inhibited the growth of LAB, which lacks fermentable substrates after aerobic exposure [35]. The number of LAB, after 7 d aerobic exposure, was only detected in 6% and 12% AA, and undesirable bacteria including yeast were also maintained at the lowest levels in 12% AA (*p* < 0.05). This was also in line with the aforementioned lower pH and higher lactic and acetic acid content in 12% AA during aerobic exposure. In addition, several studies have proved that AA had strong inhibitory effect on bacteria and fungi involve *Aspergillus flavus*, *Escherichia coli* and *Colletotrichum fragariae* [20,36]. Thus, 12% AA used in the present study can inhibit the undesirable bacteria and thereby reduce excessive spoilage of WCCS at aerobic exposure stage. 

When secondary fermentation occurred in WCCS, the richness of bacteria decreased dramatically in the CK, which was similar to the findings of Zhang et al. [37]. The lower diversity was likely because the relatively lower pH values in inoculated silages, combining activities of acidification and antagonistic activity towards other bacteria by AA, this change promotes the reduction in bacterial and fungi diversities, and ultimately improved feed quality. This observation indicated that ZA3 and AA can change fungi community when environment changes from anaerobic to aerobic. It is possible that acid-tolerant bacteria still dominate the bacterial community in the early period of aerobic exposure, and the variation of microbial community might explain the difference in fermentation quality [15,38,39]. 

Final feed quality is largely influenced by the species and numbers of dominant microorganisms in the fermentation process [40]. Firmicutes were still the dominant phylum in all treated groups after aerobic exposure 3 d, which is in accordance with the report of Zhang et al. [37]. Firmicutes are vital acid hydrolytic microbes under anaerobic condition, and acidic and anaerobic environment could produce acid and varieties of enzymes, which was conducive to the growth of Firmicutes [41]. The majority of bacteria involved in lactic acid fermentation belong to the Firmicutes. However, once the WCCS silage was exposed to oxygen, Firmicutes were gradually replaced by Proteobacteria, which might be because an increase in pH and decrease in organic acids are not beneficial to the proliferation of Firmicutes [15,39]. 

*Enterobacter* is generally considered to be undesirable during fermentation because they can ferment lactic acid to acetic acid, succinic acid and some endotoxins, which can cause degradation of fermentation quality and feed contamination. In the present study, *Enterobacter*, positively with pH and NH_3_-N, and negative lactic acid and WSC, was not detected in 6% and 12% AA after aerobic exposure. This should also be one of the reasons fermentation quality remained well in these two groups. After aerobic exposure 7 d, *Bacillus* increased by adding AA, and *Bacillus* in all AA treated groups were significantly higher, it was even positive with acetic acid. For *Bacillus* species that could produce antifungi and bacteriocin to inhibit pathogens and improve the aerobic stability of silage [42,43], AA treated groups maintained good aerobic stability. 

*Kazachstania*, *Candida* and *Pichia* were dominated in WCCS after 60 d fermentation and aerobic exposure, and a similar study was performed by Santos et al. [44], who reported that the dominant genera of yeast in WCCS were *Candida*, *Pichia*, and *Kazachstania*. After 60 d of silage, the relative abundances of *Kazachstania* were lower in 6% and 12% AA; additionally, while aerobic exposure 3 and 7 d, *Kazachstania* in 6% and 12% AA were all lower than that in other groups, *Pichia* in 6% AA+ZA3 decreased, and *Candida* was the highest in CK and was the lowest in 12% AA. *Kazachstania*, *Candida* and *Pichia*, which grow rapidly and increased the pH of the environment when exposed to air, accelerated undesirable microorganisms growth, and were the most commonly detected yeasts in aerobic spoiled fermented feed [37]; moreover, they might play an important role in aerobic deterioration of silage [41,45,46]. The addition of AA inhibited these fungi and has the potential to improve aerobic stability of silage.

*Pediococcus*, *Weissella* and *Lactococcus* were sensitive to lower pH, and were outcompeted by *Lactobacilli* at acidic environment [8]. *Lactobacilli* usually dominate and convert plant carbohydrate into lactic acid at the late fermentation stage, which inhibit the growth of *Enterobacteria* at the late fermentation stage, thereby, the reduction of NH_3_-N in ZA3 or AA silages was mainly attributed to the inhibition of microbial activity in the present study, and this could explain well the positively correlation among NH_3_-N, WSC and microorganisms mentioned above. Furthermore, addition of ZA3 in both 6% and 12% AA+ZA3 led to lower relative abundance of *Lactobacilli* compared to groups only treated with ZA3, 6% and 12% AA, which was also consistent with the results in the incubation of microbial population by plate count method (Table 3). This should be because AA contains varieties of active compounds such as essential oils and flavonoids, which have strong broad-spectrum antibacterial effects [20,21], combined with the dual action of ZA3, thus, both undesirable microorganisms and *Lactobacilli* were inhibited during 60 d fermentation and the whole aerobic exposure phase. The effect on the abundance of microbial populations as expressed decreased relative abundance of *Lactobacilli* in 6% and 12% AA+ZA3 groups. All the above findings suggest that 6% and 12% AA might be more suitable as a feed additive to inhibit fungi, especially during aerobic exposure.

Mycotoxins are secondary metabolites produced by different fungi species, and contamination of mycotoxins is widespread during feed storage [47]. AFB1, DON, ZEA, OA and FUM are the main mycotoxins of concern in WCCS [48]. Biodegradation with microorganisms and Chinese herbal medicine might be an effective method to eliminate mycotoxins [15,49]. Although LAB are effective in hindering mold growth, increasing in the oxygen concentration could provide the adequate growth conditions for these fungi strains, which would metabolize lactic acid and WSC into CO_2_ and water in aerobic condition [50]. pH dropped rapidly during 60 d of WCCS in this study, inhibited the growth of toxigenic fungi, therefore reduced the production of mycotoxin in silage, which is consistent with the research of Mugabe et al. [51], who reported that the silage process can be rapidly decrease pH and reduce the counts of yeast. After aerobic exposure for 3 d, all 5 mycotoxins in the CK were increased even compared with 60 d of fermentation; while after aerobic 7 d, those in CK again reached a higher level and all exceeded the maximum content in feed, in contrast, they were all lower than the maximum content in groups 6% and 12% AA. This also echoed that the fungi in these two groups was significantly inhibited, especially during aerobic exposure, as structural and quantitative changes of fungi communities were also closely related to mycotoxins [12]. 

A number of studies have shown that flavonoids have a certain inhibitory effect on Gram-negative and positive bacteria and fungi, and the mechanism of action is probably due to the fact that flavonoids are discretionary derivatives, which can cause the cell walls and cell membranes to break down, the cell contents to diffuse, the permeability of cell membranes to change, and the imbalance of material and information transfer inside and outside the cells to inhibit the growth of microorganisms [52,53]. This also explains why mold and yeast populations were significantly inhibited and contents of mycotoxins were lower during aerobic exposure stages in AA groups, which had significantly higher TF contents in these groups from 60 d fermentation to the whole aerobic exposure as for this study. Furthermore, TF that was negatively with *Acetobacter*, *Enterobacteria*, *Kazachstania* and *Pichia*, also the same as reports that TF were effective antibacterial against undesirable microorganisms such as *Aeromonas*, *Staphylococcus aureus*, *Escherichia coli*, *Fusarium*, *Aspergillus* and some yeasts [54,55,56,57,58]. In conclusion, TF from AA inhibit undesirable microorganisms to improve the quality of fermented feed during the fermentation and aerobic exposure process, and the results also provide further evidence that rarely reported AA can be used as additives in fermented feed.

## 4. Materials and Methods

### 4.1. Materials Collection and Analysis

On September 2020, whole crop corn (WCC, cultivated variety, Xundan 20) was harvested at milk stage in Yuzhou, China (33.76° N, 113.21° E), and whole crop AA (cultivated variety, Tangyin Beiai) was obtained at the second crop in Anyang, China (35.92^◦^ N, 114.35^◦^ E). To minimize the effect of long-distance that would affect the bioactive compounds, WSC content and LAB counts on fresh material, AA and corn were collected from different places at almost same time, both with sterile gloves and samples, and were immediately transported to the laboratory by cold chain at 4 °C, which took no more than 3 h from sampling to silage preparation and fresh sample analysis. All materials were cut to 1–2 cm for use.

Took 10 g of WCC and AA samples suspended with 90 mL distilled water and then filtered to determine pH value and ammonia nitrogen (NH_3_-N, % Total Nitrogen (TN)), respectively. The pH value and NH_3_-N concentration were measured using pH meter (Mettler Toledo Co., Ltd., Greifensee, Switzerland) and phenol-hypochloric acid colorimetry provided by Broderick et al. [59], respectively.

DM contents of WCC and AA were determined after dried at 65 °C for 48 h, and the dried samples were milled to pass through a 1 mm screen with a laboratory knife mill and used for determination of WSC, CP, EE, NDF and ADF. The WSC was measured by anthrone colorimetry using spectrophotometer (UV mini-1240, Shimadzu, Tokyo, Japan) [60]. The CP content was determined by DigiPREP TKN Systems (UDK159, Velp, Italy) and EE was by SER148 EE analyzer (SER148, Vely, Italy) according to standard procedures detailed by the Association of Official Analytical Chemists [61]. The NDF and ADF contents was analyzed according to Van Soest et al. [62] using FIWE3/6 CF analyzer (FIWE3/6, Velp, Italy).

Microbiological analysis referred to the method of Pang et al. [63] with some modifications. Then, 10 g of WCC and AA samples were diluted with 90 mL of distilled water, respectively, the supernatant was diluted serially to 10-fold and inoculated in triplicate on different agar plates: (1) LAB were measured by plate count on de MRS agar incubated at 37 °C for 48 h under anaerobic incubator; (2) Potato Dextrose Aga (PDA, containing 0.15% of tartaric acid) incubated at 37 °C for 60 h to enumerate yeast and mold; (3) and (4) Eosin Methylene Blue (EMB) agar and Nutrient Agar (NA) at 37 °C for 48 h to enumerate coliform bacteria and aerobic bacteria; (5) after incubation for 15 min in a 75 °C water bath, 10^−1^ and 10^−2^ was spread on NA and *Clostridium* enrichment medium (CLO) agar for bacillus and *Clostridium*, respectively. The colonies were counted as the numbers of viable microorganisms in cfu/g of fresh matter (FM).

### 4.2. Lactic Acid Bacteria and Silage Preparation

*Lactiplantibacillus* (*L*.) *plantarum* subsp. *plantarum* ZA3 isolated from healthy weaned piglets fecal with broad-spectrum activity and kept in our lab was used as inoculant [64]. Single colonies ZA3 was cultured in Man, Rogosa, Sharpe (MRS) medium at 37 °C for 12 h, then, centrifuged the culture at 12,000 g for 10 min at 4 °C and mixed precipitate with distilled water to make OD_600_ to 1.0, and ZA3 was 2% at 1×10^8^ cfu/mL.

Experimental treatments were designed as follows: (1) Control (CK); (2) ZA3: ZA3+CK; (3) 6% AA: 6% AA+CK; (4) 12% AA: 12% AA+CK; (5) 6% AA+ZA3: 6% AA+ZA3+CK; (6) 12% AA+ZA3: 12% AA+ZA3+CK. Then, 500 g of material was mixed homogenously separate in the above proportions and packed manually into plastic film bags (dimensions, 200 mm × 300 mm; Dragon N-6; Asahi Kasei Co., Tokyo, Japan), vacuumed, and sealed with a vacuum sealer (P-290, Shineye, Dongguan, China). A total of 72 bags (3 times × 6 treatments × 4 replicates) were prepared and kept at environmental temperature (7–36 °C) for 60 d, and then aerobic exposure was carried out for 3 and 7 d.

### 4.3. Fermentation Quality, Microbial Population, and Chemical Composition of Silage

At each opening, 3 bags of each group were randomly selected and opened for fermentation quality, microbial population and chemical composition analyzing. The concentrations of organic acids were measured using a high-performance liquid chromatography (HPLC) (Waters Alliance e2695, Waters, MA, USA) (column: Carbomix H-NP10: 8%, 7.8 × 300 mm, Sepax Technologies, Inc., Newark, DE, USA; detector: Diode Array Detector (DAD), 214 nm, Agilent 1200 Series, Agilent Technologies Co., MNC, Santa Clara, CA, USA; eluent: 2.5 mmol/L H_2_SO_4_, 0.6 mL/min; temperature: 55 °C) according to Wang et al. [64]. Limits of detection (LOD) and limits of quantification (LOQ) were 0.36 and 10 µL, respectively.

The pH value, NH_3_-N content, chemical composition and microbial population analyses were same as methods described in Section 2.1.

### 4.4. Bacterial and Fungi Community Analyses

WCCS fermented 60 d, aerobic exposure of 3 d and 7 d were sampled to analysis the bacterial and fungi communities. Briefly, 10 g of each sample were shaken well with 90 mL of sterile phosphate buffer saline at 150 rpm for 60 min, and then filtered through 4 layers of gauze, and the solution was centrifuged for 10 min at 8000 g to collect microorganism cells for DNA extraction [15]. Total genome DNA from samples was extracted using CTAB/SDS method, and DNA concentration and purity was monitored on 1% agarose gels. According to the concentration, DNA was diluted to 1 ng/μL using sterile water. The bacterial 16S rDNA (16S V4) were amplified using primers 515F (5′-GTGCCAGCMGCCGCGGTAA-3′) and reverse primer 806R (5′-GGACTACHVGGG TWTCTAAT-3′), fungi ITS was amplified using the ITS1F (5′-CTTGGTCATTTAGAGG AAGTAA-3′) and ITS2-2043R (5′-GCTGCGTTCTTCATCGATGC-3′). All PCR reactions were carried out with Phusion^®^ High-Fidelity PCR Master Mix (New England Biolabs, Inc., Beijing, China), and PCR products were detected by 2% agarose gel electrophoresis. Samples with bright main strip between 400–450 bp were chosen and purified with Qiagen Gel Extraction Kit (Qiagen, Dusseldorf, Germany). 

Sequencing libraries were generated using TruSeq^®^ DNA PCR Sample Preparation Kit (Illumina, CA, USA) following manufacturer’s recommendations and index codes were added, libraries quality was assessed on the Qubit@ 2.0 Fluorometer (Thermo Fisher Scientific Inc., Carlsbad, CA, USA) and Agilent Bioanalyzer 2100 system (Agilent Technologies, Santa Clara, CA, USA), and libraries were sequenced on IlluminaHiSeq2500 platform (Shanghai Applied Protein Technology Co., Ltd., Shanghai, China) and 250 bp paired-end reads were generated. Sequences analysis were performed by Uparse software (Uparse v7.0.1001, http://drive5.com/uparse/accessed on 3 November 2020), and sequences with 97% similarities were assigned to the same operational taxonomic units (OTUs). Alpha diversity was applied in analyzing complexity of species diversity for a sample through Chao1 and Shannon using QIIME (Version 1.7.0) and displayed with R software (Version 2.15.3, http://www.mothur.org/wiki/Chao1 and http://www.mothur.org/wiki/Shannon accessed on 24 July 2021). 

PCoA analysis was demonstrated for the variance of the bacterial and fungi community structure by weighted gene co-expression network analysis (WGCNA), stat and ggplot2 packages in R software, and Spearman correlation heatmap based on the Spearman correlation coefficients among the bacterial and fungi community and fermentation parameters were produced using R software (version 2.15.3). 

### 4.5. Mycotoxin Determination

Mycotoxin including AFB1, DON, ZEN, OA and FUM were analyzed using enzyme linked immunosorbent assay (ELISA) kits provided by Lianshuo Biological Technology Co., Ltd. (AMEKO, Shanghai, China).

### 4.6. Total Flavonoid Measurement

The total flavonoid (TF) content was determined using the slightly modified procedure of spectrometry provided by Hudz et al. [65] and made calibration curves rutin. The absorbance of the reaction mixtures was measured at OD_510_.

### 4.7. Statistical Analyses

The data of fermentation characteristics, microbial population, chemical composition, mycotoxin and TF were subjected using the IBM SPSS statistical package 22.0 (SPSS Inc., Chicago, Illinois, IL, USA), and Student–Newman–Keuls multiple range tests were used to evaluate differences among treatments and the significance was declared at *p* < 0.05. The sequencing data were analyzed using the free online Lingbo Microclass (http://www.biomicroclass.com accessed on 25 November 2021). Correlation analyses of microbial and fermentation qualities, TF and microbial, were performed using Spearman’s rank correlation coefficient and Pearson’s rank correlation coefficient, respectively, and each experiment was performed in triplicate.

## 5. Conclusions

This research found that fermentation quality, chemical composition and microbial population of whole crop corn silage could be improved by treating with *Lactiplantibacillus plantarum* subsp. *plantarum* ZA3 and *Artemisia argyi*. After 60 d of fermentation, both ZA3 and higher application level of AA (> 6%) increased the relative abundances of *Lactobacilli*, moreover, AA decreased *Enterobacter* and *Acetobacter*. During aerobic exposure, for the relative abundances of microbiology, ZA3 and AA lowered abundances of *Candida*, while ZA3 also reduced *Acetobacter*, AA decreased *Pichia* and *Kazachstania*; additionally, levels of DON, OA and FUM were reduced by ZA3 and 12% AA, besides, 12% AA had effect on degrading AFB1 and ZEN. Total flavonoid content was significantly higher in all AA treated groups from 60 d fermentation to the whole aerobic exposure; in addition, it was positively correlated with *Paenibacillus*, *Weissella* and *Lactobacilli*, and negatively with *Acetobacter*, *Enterobacteria*, *Kazachstania* and *Pichia*. The results of this research provide a preliminary reference for the possibility of applying AA (> 6%) as efficient feed additives to inhibit mycotoxin derived by fungi and improve silage quality.

## Figures and Tables

**Figure 1 toxins-14-00349-f001:**
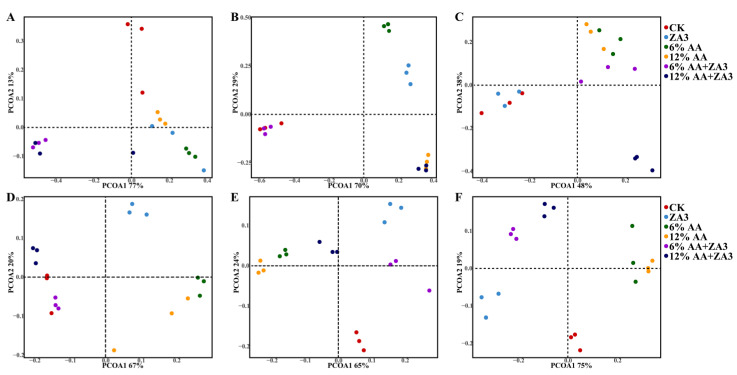
Principal coordinates analysis on genus level of bacterial and fungi community during ensiling and aerobic exposure. Bacterial community after 60 d of ensiling (**A**), bacterial community after aerobic exposure for 3 d (**B**) and bacterial community after aerobic exposure for 7 d (**C**). Fungi community after 60 d of ensiling (**D**), fungi community after aerobic exposure for 3 d (**E**) and fungi community after aerobic exposure for 7 d (**F**).

**Figure 2 toxins-14-00349-f002:**
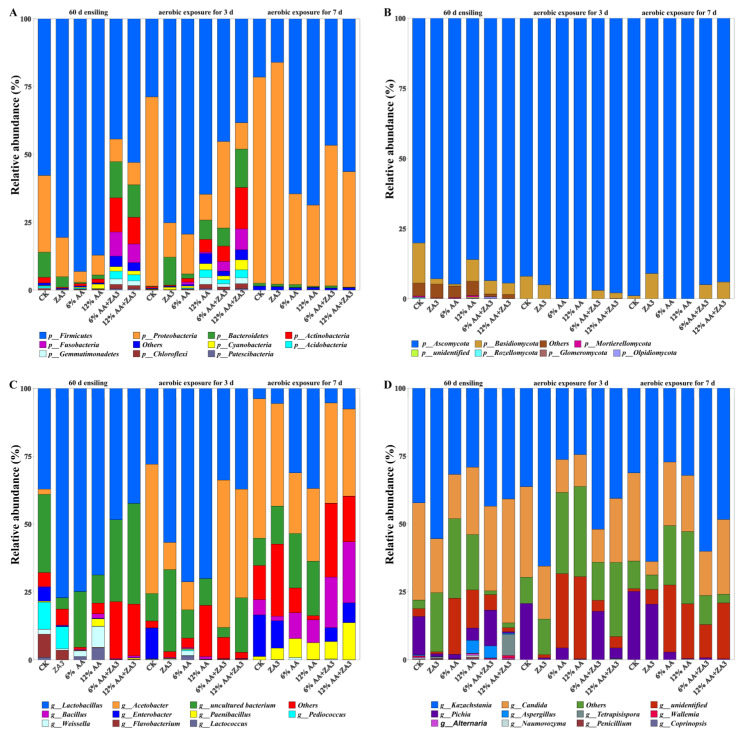
The bacterial and fungi community of WCCS after 60 d of ensiling and aerobic exposure. The bacterial communities are shown at the phylum level (**A**) and the genus level (**C**). The fungi communities are shown at the phylum level (**B**) and the genus level (**D**).

**Figure 3 toxins-14-00349-f003:**
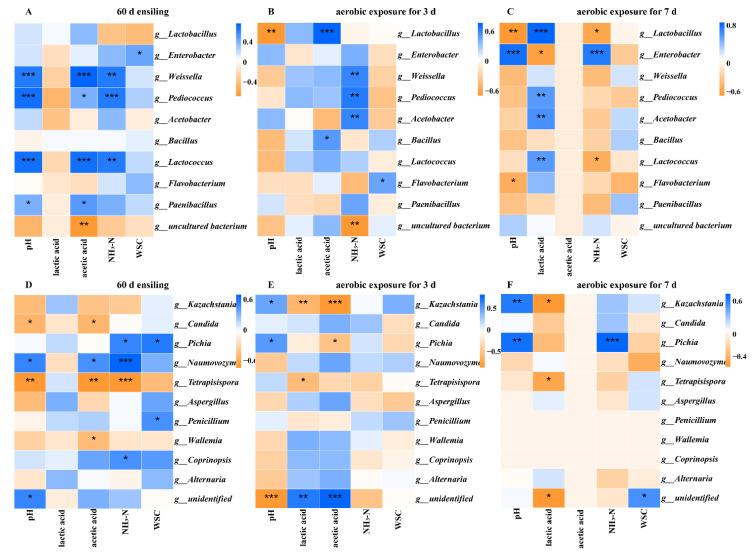
Spearman correlation heatmap of abundance of the top 10 enriched bacteria and fungi at the genus level with fermentation properties during fermentation and aerobic exposure. Bacterial: 60 d ensiling (**A**), 3 d (**B**) and 7 d (**C**) of aerobic exposure; Fungi: 60 d ensiling (**D**), 3 d (**E**) and 7 d (**F**) of aerobic exposure. Positive correlations are shown in blue, and negative correlations are shown in orange. * *p* < 0.05; ** *p* < 0.01; *** *p* < 0.001. d, days.

**Figure 4 toxins-14-00349-f004:**
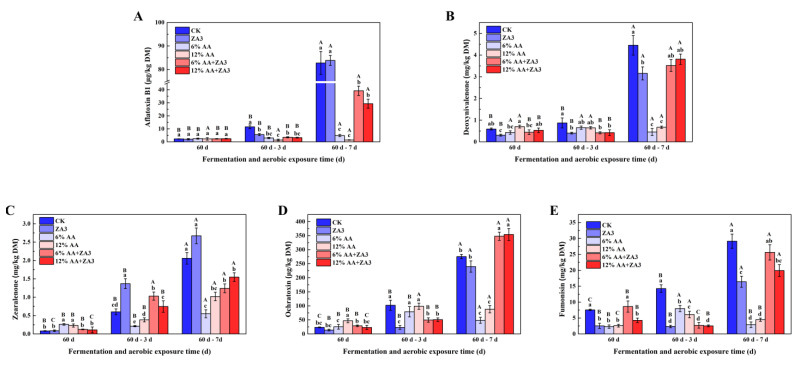
Content of toxin after 60 d of ensiling and during the aerobic exposure. Aflatoxin B1 (μg/kg DM) (**A**), Deoxynivalenone (mg/kg DM) (**B**), Zearalenone (mg/kg DM) (**C**), Fumonisin (mg/kg DM) (**D**), Ochratoxin (μg/kg DM) (**E**). Different lowercase letters (a–d) for the same treatment time indicate significant differences (*p* < 0.05) among different treatment groups. Different capital letters (A–C) for the same treatment groups indicate significant differences among different treatment times (*p* < 0.05) according to Student–Newman–Keuls test.

**Figure 5 toxins-14-00349-f005:**
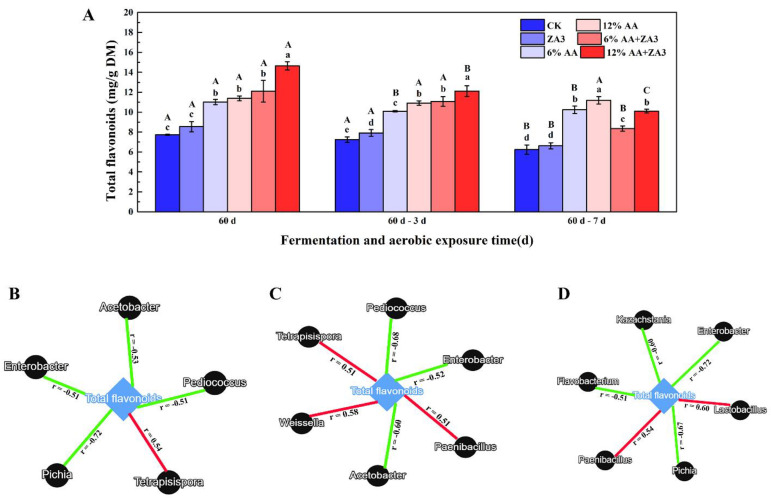
Content of total flavonoid (TF) and linkages with microbial during fermentation and aerobic exposure. (**A**), TF (mg/g DM). (**B**–**D**), co-occurrence networks analysis the top 10 enriched bacteria and fungi at the genus level with TF in 60 d of ensiling, and 60 d-3 d and 60 d-7 d of aerobic exposure. Different lowercase letters (a–e) for the same treatment time indicate significant differences (*p* < 0.05) among different treatment groups. Different capital letters (A–C) for the same treatment groups indicate significant differences among different treatment times (*p* < 0.05) according to Student–Newman–Keuls test. d, days. Each co-occurrence pair among top 10 enriched bacteria and fungi at the genus level and TF has an absolute Pearson rank correlation above 0.50 (red straight line, positive correlation (r ≥ 0.50); green straight line, negative correlation (r ≤ −0.50)). Microbes are shown by black circle shaped nodes, and TF is shown by blue rhombus nodes.

**Table 1 toxins-14-00349-t001:** Characteristics and microbial population of WCC and AA (Mean ± SD, *n* = 3).

Item	WCC	AA
pH	6.61 ± 0.05	5.58 ± 0.01
DM (g/kg)	277.03 ± 3.29	371.55 ± 3.77
WSC (% DM)	12.12 ± 0.22	5.41 ± 0.25
CP (% DM)	9.43 ± 0.31	10.92 ± 0.02
NDF (% DM)	60.07 ± 0.56	53.76 ± 0.84
ADF (% DM)	35.66 ± 0.78	30.42 ± 0.51
EE (% DM)	2.51 ± 0.06	3.06 ± 0.08
Microbial population (lg cfu/g FM)		
LAB	6.33 ± 0.17	5.59 ± 0.08
Aerobic bacteria	8.32 ± 0.24	5.84 ± 0.05
Coliform bacteria	7.81 ± 0.15	5.63 ± 0.46
Yeast	5.76 ± 0.09	ND
Bacilli	4.68 ± 0.13	6.61 ± 0.01

SD, standard deviation; DM, dry matter; WSC, water-soluble carbohydrates; CP, crude protein; NDF, neutral detergent fiber; ADF, acid detergent fiber; EE, ether extract; LAB, lactic acid bacteria; cfu, colony forming unit; FM, fresh material.

**Table 2 toxins-14-00349-t002:** Fermentation quality of WCCS during ensiling and aerobic exposure (Mean ± SD, *n* = 3).

Item	Treatment	pH	NH_3_-N (% TN)	Organic Acid (g/Kg DM)		WSC (%DM)
				Lactic Acid	Acetic Acid	
60 d ensiling	CK	3.73 ± 0.02 ^bC^	16.11 ± 0.23 ^aB^	25.96 ± 0.97 ^bA^	3.16 ± 0.3 ^b^	1.51 ± 0.01 ^aA^
	ZA3	3.73 ± 0.01 ^cC^	15.42 ± 0.11 ^aB^	32.90 ± 1.57 ^aA^	3.39 ± 0.33 ^ab^	1.43 ± 0.01 ^aA^
	6% AA	3.75 ± 0.01 ^aC^	14.63 ± 0.41 ^abA^	26.40 ± 1.67 ^bA^	3.57 ± 0.38 ^ab^	1.45 ± 0.02 ^aB^
	12% AA	3.70 ± 0.01 ^bB^	19.21 ± 0.22 ^bA^	32.44 ± 1.24 ^aA^	3.87 ± 0.25 ^aB^	1.72 ± 0.03 ^aA^
	6% AA+ZA3	3.52 ± 0.02 ^dC^	12.42 ± 0.64 ^bcA^	33.18 ± 0.40 ^aA^	ND	1.44 ± 0.01 ^aA^
	12% AA+ZA3	3.52 ± 0.01 ^dC^	12.07 ± 0.51 ^cAB^	27.05 ± 1.60 ^bA^	ND	1.09 ± 0.01 ^bC^
aerobic exposure for 3 d	CK	5.14 ± 0.09 ^aB^	21.12 ± 0.33 ^aC^	9.71 ± 0.69 ^cB^	ND	1.16 ± 0.01 ^aB^
	ZA3	4.34 ± 0.1 ^bB^	19.06 ± 0.52 ^bB^	8.58 ± 1.37 ^cB^	ND	1.61 ± 0.02 ^aA^
	6% AA	3.90 ± 0.02 ^dB^	17.56 ± 0.22 ^bcA^	19.29 ± 4.45 ^bB^	ND	1.16 ± 0.03 ^aB^
	12% AA	3.70 ± 0.03 ^eB^	13.17 ± 0.21 ^cB^	41.45 ± 8.66 ^aA^	6.52 ± 1.53 ^aA^	1.27 ± 0.02 ^aA^
	6% AA+ZA3	4.19 ± 0.1 ^cB^	13.63 ± 0.63 ^cB^	17.14 ± 1.53 ^bB^	ND	1.29 ± 0.01 ^aA^
	12% AA+ZA3	4.44 ± 0.03 ^bB^	14.22 ± 0.92 ^cB^	4.67 ± 0.16 ^dB^	ND	1.35 ± 0.01 ^aB^
aerobic exposure for 7 d	CK	7.74 ± 0.1 ^aA^	41.09 ± 0.41 ^aA^	ND	ND	1.63 ± 0.04 ^aA^
	ZA3	6.98 ± 0.06 ^bA^	32.51 ± 0.63 ^bA^	ND	ND	1.48 ± 0.03 ^aA^
	6% AA	6.48 ± 0.1 ^cA^	24.32 ± 0.79 ^bcA^	4.56 ± 0.44 ^bC^	ND	1.11 ± 0.07 ^bA^
	12% AA	4.64 ± 0.16 ^dA^	19.93 ± 0.18 ^cB^	7.48 ± 1.71 ^aB^	ND	0.50 ± 0.01 ^cB^
	6% AA+ZA3	6.66 ± 0.01 ^cA^	24.47 ± 0.74 ^bcA^	ND	ND	1.63 ± 0.05 ^aA^
	12% AA+ZA3	6.54 ± 0.04 ^cA^	22.29 ± 0.52 ^bcA^	ND	ND	1.23 ± 0.04 ^bA^
SEM		0.01	0.006	0.342	0.053	0.019
Period		***	***	***	***	**
Treatment		***	***	***	***	**
Interaction		***	***	***	***	***

DM, dry matter; NH_3_-N, ammonia nitrogen; WSC, water-soluble carbohydrates; TN, total nitrogen; ND, not detected; Different lowercase letters (^a–d^) mean the same treatment time indicate significant differences (*p* < 0.05) among different treatment groups. Different capital letters (^A–C^) mean the same treatment groups indicate significant differences among different treatment times (*p* < 0.05). The significant difference between period and treatment inter-actions is expressed as ** *p* < 0.01 and *** *p* < 0.001.

**Table 3 toxins-14-00349-t003:** Chemical composition of WCCS during ensiling and aerobic exposure (Mean ± SD, *n* = 3).

Item	Treatment	Chemical Composition (%DM)				
		CP	NDF	ADF	EE	DM
60 d ensiling	CK	9.85 ± 0.34 ^aA^	60.75 ± 0.62 ^aA^	38.15 ± 0.67 ^aC^	2.59 ± 0.02 ^cA^	212.08 ± 6.47 ^cA^
	ZA3	9.60 ± 0.14 ^aA^	59.05 ± 0.79 ^abA^	36.95 ± 0.58 ^abB^	2.77 ± 0.05 ^dA^	211.42 ± 7.44 ^cB^
	6% AA	9.67 ± 0.17 ^aA^	56.08 ± 0.98 ^cA^	35.86 ± 0.10 ^bcB^	2.93 ± 0.06 ^cAB^	267.26 ± 6.86 ^bB^
	12% AA	10.11 ± 0.10 ^aA^	56.33 ± 1.01 ^cA^	36.11 ± 0.62 ^bcB^	3.04 ± 0.12 ^abB^	294.29 ± 7.97 ^aA^
	6% AA+ZA3	9.72 ± 0.28 ^aA^	57.79 ± 0.91 ^bcA^	34.86 ± 0.71 ^cB^	2.92 ± 0.08 ^cB^	256.91 ± 6.22 ^bA^
	12% AA+ZA3	9.76 ± 0.21 ^aA^	58.00 ± 0.52 ^bcA^	35.48 ± 0.27 ^bcB^	3.16 ± 0.06 ^aA^	301.87 ± 9.71 ^aA^
aerobic exposure for 3 d	CK	9.04 ± 0.07 ^bB^	59.26 ± 1.13 ^aA^	37.76 ± 0.61 ^abB^	2.53 ± 0.01 ^eA^	220.99 ± 4.83 ^cA^
	ZA3	9.64 ± 0.02 ^abA^	59.08 ± 0.11 ^aA^	37.85 ± 1.24 ^aB^	2.76 ± 0.14 ^dA^	228.26 ± 6.91 ^cA^
	6% AA	9.76 ± 0.12 ^aA^	56.65 ± 0.74 ^bA^	35.42 ± 1.21 ^cB^	2.84 ± 0.13 ^cdB^	274.87 ± 4.02 ^bB^
	12% AA	10.06 ± 0.43 ^aA^	56.60 ± 0.49 ^bA^	35.31 ± 0.29 ^cB^	2.99 ± 0.07 ^bcB^	305.32 ± 6.99 ^aA^
	6% AA+ZA3	9.47 ± 0.45 ^abA^	58.37 ± 0.66 ^abA^	35.47 ± 0.43 ^bcB^	3.13 ± 0.07 ^abA^	268.64 ± 4.06 ^bA^
	12% AA+ZA3	10.03 ± 0.27 ^aA^	57.86 ± 0.85 ^abA^	36.15 ± 0.76 ^abcB^	3.28 ± 0.11 ^aA^	303.05 ± 3.97 ^aA^
aerobic exposure for 7 d	CK	8.92 ± 0.08 ^cB^	58.53 ± 0.74 ^aB^	39.46 ± 0.68 ^aA^	2.49 ± 0.10 ^bA^	209.39 ± 1.07 ^dA^
	ZA3	8.98 ± 0.23 ^bcB^	58.41 ± 0.93 ^aB^	39.65 ± 0.49 ^aA^	2.53 ± 0.11 ^bA^	210.55 ± 5.88 ^dB^
	6% AA	9.31 ± 0.07 ^bcB^	56.06 ± 0.48 ^bA^	37.94 ± 0.87 ^abA^	3.14 ± 0.11 ^aA^	293.97 ± 6.94 ^bA^
	12% AA	10.05 ± 0.07 ^aA^	56.01 ± 0.88 ^bA^	37.33 ± 0.68 ^bA^	3.24 ± 0.02 ^aA^	318.23 ± 11.12 ^aA^
	6% AA+ZA3	9.09 ± 0.19 ^bcA^	56.47 ± 1.17 ^abB^	38.51 ± 0.40 ^aA^	3.15 ± 0.11 ^aA^	277.24 ± 3.84 ^cA^
	12% AA+ZA3	9.40 ± 0.31 ^bA^	56.73 ± 0.39 ^abB^	38.35 ± 0.70 ^aA^	3.29 ± 0.11 ^aA^	309.02 ± 7.94 ^abA^
SEM		0.03	0.106	0.091	0.012	0.172
Period		**	*	**	*	*
Treatment		***	**	**	***	***
Interaction		**	**	**	***	***

CP, crude protein; NDF, neutral detergent fiber; ADF, acid detergent fiber; EE, ether extract; DM, dry matter; Different lowercase letters (^a–d^) mean the same treatment time indicate significant differences (*p* < 0.05) among different treatment groups. Different capital letters (^A–C^) mean the same treatment groups indicate significant differences among different treatment times (*p* < 0.05). The significant difference between period and treatment inter-actions is expressed as * *p* < 0.05, ** *p* < 0.01 and *** *p* < 0.001.

**Table 4 toxins-14-00349-t004:** Microbial population of WCCS during ensiling and aerobic exposure (Mean ± SD, *n* = 3).

Item	Treatment	Microbial Population (lg cfu/g FM)				
		Lactic Acid Bacteria	Aerobic Bacteria	Coliform Bacteria	Yeast	Bacilli
60 d ensiling	CK	6.97 ± 0.05 ^cA^	5.64 ± 0.21 ^aC^	3.96 ± 0.24 ^bC^	5.11 ± 0.14 ^aB^	3.70 ± 0.01 ^aC^
	ZA3	8.02 ± 0.03 ^aA^	5.36 ± 0.30 ^aC^	4.16 ± 0.41 ^bB^	4.25 ± 0.52 ^bC^	ND
	6% AA	6.83 ± 0.13 ^cA^	5.31 ± 0.11 ^aC^	4.28 ± 0.27 ^bC^	4.88 ± 0.07 ^aC^	4.44 ± 0.02 ^bC^
	12% AA	7.48 ± 0.06 ^bA^	5.68 ± 0.05 ^aB^	4.20 ± 0.17 ^bC^	4.76 ± 0.19 ^aC^	4.00 ± 0.13 ^cB^
	6% AA+ZA3	6.51 ± 0.06 ^cA^	5.75 ± 0.14 ^aC^	4.99 ± 0.16 ^aC^	5.09 ± 0.14 ^aC^	4.57 ± 0.02 ^bC^
	12% AA+ZA3	6.35 ± 0.07 ^cA^	5.65 ± 0.12 ^aC^	5.10 ± 0.12 ^aC^	4.90 ± 0.19 ^aC^	5.39 ± 0.11 ^aB^
aerobic exposure for 3 d	CK	4.27 ± 0.11 ^dB^	8.14 ± 0.14 ^aB^	7.81 ± 0.12 ^aB^	10.17 ± 0.12 ^aA^	5.10 ± 0.06 ^aB^
	ZA3	6.02 ± 0.15 ^bB^	7.36 ± 0.12 ^bcB^	4.56 ± 0.23 ^bB^	9.28 ± 0.14 ^bB^	3.90 ± 0.12 ^cB^
	6% AA	6.03 ± 0.10 ^bA^	7.24 ± 0.27 ^cB^	5.10 ± 0.40 ^bB^	9.20 ± 0.26 ^bB^	4.93 ± 0.17 ^aB^
	12% AA	7.12 ± 0.09 ^aA^	4.93 ± 0.11 ^dB^	5.00 ± 0.27 ^bB^	7.61 ± 0.33 ^cB^	3.70 ± 0.13 ^cC^
	6% AA+ZA3	5.03 ± 0.07 ^cB^	7.40 ± 0.05 ^bcB^	7.34 ± 0.31 ^aB^	9.49 ± 0.06 ^bB^	5.16 ± 0.04 ^aB^
	12% AA+ZA3	4.93 ± 0.04 ^cB^	7.68 ± 0.10 ^bB^	7.39 ± 0.26 ^aB^	9.39 ± 0.14 ^bB^	4.71 ± 0.14 ^bC^
aerobic exposure for 7 d	CK	ND	10.09 ± 0.04 ^aA^	10.13 ± 0.08 ^aA^	10.20 ± 0.15 ^aA^	7.45 ± 0.21 ^aA^
	ZA3	ND	10.18 ± 0.01 ^aA^	9.79 ± 0.16 ^bA^	9.94 ± 0.04 ^bA^	7.36 ± 0.32 ^abA^
	6% AA	3.30 ± 0.15 ^bB^	9.82 ± 0.18 ^bA^	9.57 ± 0.05 ^bcA^	9.85 ± 0.05 ^bA^	7.31 ± 0.07 ^abA^
	12% AA	4.73 ± 0.09 ^aB^	9.46 ± 0.11 ^cA^	9.09 ± 0.14 ^dA^	9.20 ± 0.20 ^cA^	6.32 ± 0.16 ^cA^
	6% AA+ZA3	ND	9.62 ± 0.26 ^bcA^	9.59 ± 0.21 ^bcA^	9.86 ± 0.20 ^bA^	7.26 ± 0.11 ^abA^
	12% AA+ZA3	ND	9.75 ± 0.20 ^bA^	9.47 ± 0.08 ^cA^	9.78 ± 0.08 ^bA^	7.15 ± 0.27 ^bA^
SEM		0.008	0.02	0.031	0.027	0.014
Period		***	***	***	***	***
Treatment		***	***	***	***	***
Interaction		***	***	***	***	***

FM, fresh material; ND, not detected; Different lowercase letters (^a–d^) mean the same treatment time indicate significant differences (*p* < 0.05) among different treatment groups. Different capital letters (^A–C^) mean the same treatment groups indicate significant differences among different treatment times (*p* < 0.05). The significant difference between period and treatment inter-actions is expressed as *** *p* < 0.001.

**Table 5 toxins-14-00349-t005:** Alpha diversity of bacteria and fungi of WCCS during ensiling and aerobic exposure (Mean ± SD, *n* = 3).

Item	Treatment	Bacteria		Fungi	
		Shannon	Chao1	Shannon	Chao1
60 d ensiling	CK	4.37 ± 0.18 ^aB^	2717.73 ± 168.19 ^bA^	2.50 ± 0.12 ^bA^	126.42 ± 10.99 ^bA^
	ZA3	2.97 ± 0.66 ^cB^	1543.12 ± 209.71 ^cB^	1.96 ± 0.15 ^cA^	127.65 ± 8.99 ^bA^
	6% AA	3.19 ± 0.17 ^bB^	2289.73 ± 211.50 ^bA^	1.76 ± 0.18 ^cA^	132.53 ± 12.62 ^bA^
	12% AA	3.56 ± 0.31 ^bB^	2436.86 ± 234.93 ^bA^	1.94 ± 0.04 ^cA^	130.49 ± 7.81 ^bB^
	6% AA+ZA3	4.23 ± 0.22 ^aC^	2745.72 ± 244.11 ^bB^	3.22 ± 0.07 ^aA^	160.47 ± 5.92 ^aB^
	12% AA+ZA3	4.65 ± 0.30 ^aB^	3746.15 ± 81.39 ^aA^	2.46 ± 0.17 ^bA^	164.33 ± 6.81 ^aB^
aerobic exposure for 3 d	CK	2.25 ± 0.23 ^cC^	2100.83 ± 200.52 ^bA^	1.69 ± 0.07 ^aB^	146.67 ± 5.82 ^bA^
	ZA3	2.60 ± 0.36 ^cB^	1550.55 ± 129.86 ^cB^	1.31 ± 0.15 ^bB^	142.42 ± 2.03 ^bA^
	6% AA	3.32 ± 0.56 ^bB^	1816.18 ± 159.40 ^cB^	1.35 ± 0.10 ^bB^	114.67 ± 2.52 ^cA^
	12% AA	3.73 ± 0.22 ^bB^	1886.84 ± 119.26 ^cB^	1.82 ± 0.19 ^aA^	111.33 ± 1.53 ^cA^
	6% AA+ZA3	4.31 ± 0.16 ^abB^	2791.93 ± 182.80 ^bB^	2.29 ± 0.14 ^aB^	223.94 ± 3.28 ^aA^
	12% AA+ZA3	5.16 ± 0.19 ^aA^	4732.19 ± 178.50 ^aA^	1.85 ± 0.07 ^aB^	264.45 ± 11.09 ^aA^
aerobic exposure for 7 d	CK	5.21 ± 0.28 ^aA^	2407.26 ± 154.68 ^bA^	2.36 ± 0.12 ^aA^	74.50 ± 2.50 ^bB^
	ZA3	5.34 ± 0.34 ^aA^	2249.07 ± 184.11 ^bA^	2.08 ± 0.06 ^aA^	99.50 ± 4.77 ^bB^
	6% AA	4.31 ± 0.24 ^bA^	2096.86 ± 68.13 ^cA^	1.67 ± 0.21 ^bA^	65.33 ± 2.52 ^cB^
	12% AA	4.53 ± 0.13 ^bA^	2161.91 ± 46.62 ^cA^	1.82 ± 0.03 ^bA^	54.61 ± 2.85 ^cB^
	6% AA+ZA3	5.17 ± 0.25 ^aA^	3131.24 ± 154.94 ^aA^	2.46 ± 0.12 ^aB^	127.48 ± 7.88 ^aB^
	12% AA+ZA3	5.72 ± 0.30 ^aA^	3264.59 ± 105.54 ^aA^	2.47 ± 0.18 ^aA^	141.33 ± 10.50 ^aB^

Different lowercase letters (^a–c^) mean the same treatment times indicate significant differences (*p* < 0.05) among different treatment groups. Different capital letters (^A–C^) mean the same treatment groups indicate significant differences among different treatment times (*p* < 0.05).

## Data Availability

The data generated from the study is clearly presented and discussed in the manuscript.

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
