# Peer review of "Variation of Microbial Community and Fermentation Quality in Corn Silage Treated with Lactic Acid Bacteria and Artemisia argyi during Aerobic Exposure"

_toxins, 2022, doi:10.3390/toxins14050349_

Round 1

Reviewer 1 Report

The article presents a study on the effects of lactic acid bacteria (LAB) Lactiplantibacillus (L.) plantarum subsp. plantarum and Artemisia argyi on the fermentation characteristics, microbial community and mycotoxin of whole crop corn silage during 60 days ensiling and subsequent 7 d aerobic exposure. The undertaken research problem is interesting and the novelty of the research is explained in the introduction. Overall, the title reflects the manuscript content. The publication is well organized and written. Abstract is informative and the keywords have been properly selected.

The description of the material and methods was properly prepared, the results well described and discussed. Some parts of the work require some adjustments in accordance with the following points:

Why aerobic exposure was carried out for 3 and 7 d?

The description of the performed statistical analyzes is insufficient in relation to the presented results. It is necessary to complete this part of the work (point 4.7)

The results presented in Table 5 are not sufficiently explained. They require supplementation.

Reviewer 2 Report

Variation of microbial community and fermentation quality in corn silage treated with lactic acid bacteria and Artemisia argyi during aerobic exposure.

The work is interesting.

  • I recommend the authors to read the manuscript to correct some errors in the text.

  • All tables show the results obtained with their significance but no standard deviation (SD) is reported in any result. Were the analyses not performed in triplicate? Add data.

  • For the HPLC analysis, the method used is not specified nor are the LOD and LOQ of detectability of the organic acids considered.

Reviewer 3 Report

Very interesting manuscript and very interesting scientific research. The text has some innovation features. The manuscript is well edited and described, perfectly fits the subject of the thematic range of the TOXINS magazine. That is why I think that this manuscript should be allowed to be published in this magazine.

Before publishing, the manuscript should be slightly improved. Considering the current systematics of lactobacilli (see: http://lactobacillus.ualberta.ca/), I suggest using the term "lactobacilli" instead of "Lactobacillus" where it is possible.

Round 2

Reviewer 1 Report

Thanks to the authors for improving the work as suggested.
The work as it stands is acceptable.